# Biomimetic Magnetic Particles for the Removal of Gram-Positive Bacteria and Lipoteichoic Acid

**DOI:** 10.3390/pharmaceutics14112356

**Published:** 2022-10-31

**Authors:** Bernhard Friedrich, Julia Eichermüller, Christian Bogdan, Sarah Cunningham, Holger Hackstein, Richard Strauß, Christoph Alexiou, Stefan Lyer, Rainer Tietze

**Affiliations:** 1Department of Otorhinolaryngology, Head and Neck Surgery, Section of Experimental Oncology and Nanomedicine (SEON), Else Kröner-Fresenius-Stiftung Professorship, Universitätsklinikum Erlangen, 91054 Erlangen, Germany; 2Mikrobiologisches Institut—Klinische Mikrobiologie, Immunologie und Hygiene, Universitätsklinikum Erlangen, Friedrich-Alexander-Universität (FAU) Erlangen-Nürnberg, 91054 Erlangen, Germany; 3Department of Transfusion Medicine and Hemostaseology, Universitätsklinikum Erlangen, 91054 Erlangen, Germany; 4Department of Medicine 1, Universitätsklinikum Erlangen, 91054 Erlangen, Germany; 5Department of Otorhinolaryngology, Head and Neck Surgery, Professorship for AI-Guided Nanomaterials, Universitätsklinikum Erlangen, 91054 Erlangen, Germany

**Keywords:** GP-340-derived peptides, lipoteichoic acid, superparamagnetic iron oxide nanoparticles (SPIONs), Gram-positive bacteria

## Abstract

Gram^+^ bacteria are very common in clinical medicine and responsible for a large number of infectious diseases. For example, Gram^+^ bacteria play a major role in causing bloodstream infections and sepsis. Therefore, the detection of Gram^+^ bacteria is of great importance for the diagnosis and treatment of infectious diseases. Furthermore, these bacteria are often present in biofilms that cover implants. Recent research work has mainly focused on the biologic activity and removal of Gram-negative bacteria or bacterial components such as lipopolysaccharides (LPS). In contrast, the effects of lipoteichoic acid (LTA) have been less well studied so the relevance of their removal from body fluids is possibly underestimated. To address this topic, we evaluated superparamagnetic iron oxide particles (SPION) carrying different peptides derived from the innate immune receptor (GP-340) for their ability to bind and remove Gram^+^ bacteria and LTA from different media. Our results show that, beyond *S. aureus*, effective agglutinating and removing of *S. pneumoniae* was possible. Furthermore, we were able to show for the first time that this was possible with LTA alone and that the magnetic removal of bacteria was also efficient under flow conditions. We also found that this method was able to capture *Stapyhylococcus aureus* from platelet concentrates, which can help to enhance the sensitivity of microbiological diagnostics, quality control measures, and blood product safety.

## 1. Introduction

Bacterial pathogens can cause the formation of biofilms on implants, transfusion incidences, and bloodstream infections. This can subsequently lead to sepsis and, therefore, these pathogens are an important target for diagnostic and therapeutic approaches in infectious disease medicine [1,2,3,4,5]. Gram-positive bacteria show diverse growth characteristics and antibiotic susceptibilities [6]. A very common Gram-positive bacterium causing various local and systemic infections is *Staphylococcus (S.) aureus*, which has received considerable attention in infectious disease research [7]. Another medically important Gram-positive bacterium is *Streptococcus (S.) pneumoniae*, which is a major cause of upper and lower respiratory tract infections and meningitis [8,9]. Both bacterial species exhibit structural differences in their cell wall. The cell wall of *S. pneumoniae* shows a complex composition consisting of teichoic acid and lipoteichoic acid (LTA) that contain certain repeating units decorated with phosphoryl choline residues and a high proportion of *N*-deacetylated saccharides, which does not apply to *S. aureus* and its LTA structure [10,11,12]. While *S. aureus* preferably forms clusters of bacterial cells, *S. pneumoniae* forms chains. Both species possess a thicker peptidoglycan layer than Gram-negative bacteria [6]. Myeloid immune cells, such as neutrophilic granulocytes and monocytes, recognize these cell wall components, including LTA, via their toll-like receptor (TLR) 2, which leads to the release of inflammatory cytokines [10,13,14]. Lipopolysaccharides (LPS), a component of the Gram-negative cell wall, and LTA share some biochemical and physiological properties [15]. However, compared to LPS (detected mainly by TLR4), the same molar concentration of lipoteichoic acid (LTA) causes the release of a lower amount of cytokines [16]. Nevertheless, the proinflammatory events elicited by LTA are still sufficient to cause severe disease. Therefore, LTA is of high interest as a potential therapeutic target in bloodstream infections due to Gram-positive bacteria [17].

Recently, we were successful in removing mainly Gram-negative bacteria, as well as LPS, using a biomimetic peptide derived from the innate immune receptor GP-340. This glycoprotein functions as a broad-spectrum pattern recognition receptor capable of agglutinating various microorganisms. The actual binding site consists of only a 18-mer peptide, which we ligated to the surface of APTES-coated superparamagnetic iron-oxide particles (SPIONs) [18]. SPIONs form, in combination with different interfaces, innovative approaches in experimental medicine and can be used for both diagnostic and therapeutic purposes, for example for the treatment of breast cancer or the visualization of vascular tissue [19,20,21,22].

In our current study presented here, we examined a selection of highly soluble GP-340-derived peptides (including a non-binding peptide sequence as control) for target specificity and affinity. The different peptides, which include the canonical binding sequences and a peptide with a non-binding sequence, were tested with different Gram-positive bacteria (*S. aureus* and *S. pneumoniae*) for their agglutination capability. Several peptides for further evaluation were selected and bound to SPION-APTES and the particles were characterized afterwards [18]. Magnetic separations were performed for the first time at clinically relevant concentrations of bacteria and LTA from aqueous media. Both *S. aureus* and *S. pneumoniae* were separated under static conditions, whereas *S. aureus* was also tested in a first trial flow model. Furthermore, we tried to make these findings directly applicable to detecting microbial contaminants in blood products. Accordingly, we evaluated whether intact *S. aureus* could be removed from platelet concentrates and enriched for diagnostic purposes, offering a highly efficient sample preparation method for detecting bacterial contamination in biological fluids.

## 2. Materials and Methods

### 2.1. Materials

The deionized H_2_O used in the experiments was prepared using the Merck Milli-Q^®^ direct water purification system (Darmstadt, Germany). The iron calibration standard was obtained from Bernd Kraft (Duisburg, Germany). Merck (Darmstadt, Germany) supplied iron (II) chloride tetrahydrate, iron (III) chloride hexahydrate, calcium chloride, and Tween 20. Ringer solution was obtained from Fresenius Kabi (Bad Homburg, Germany). RPMI 1640 medium supplemented with 10% fetal calf serum (FCS) and 1% L-glutamine was obtained from Live Technologies (Carlsbad, CA, USA). *N*-Succinimidyl bromoacetate was supplied by VWR International GmbH (Damstadt, Germany). Calcium chloride, boric acid, tris-(hydroxymethyl)-aminomethane, *N*,*N*-dimethylformamide, 3-aminopropyltriethoxysilan, ammonia (25% *v*/*v* solution), and hydrochloric acid (HCl) were supplied by Carl Roth GmbH (Karlsruhe, Germany).

The peptides PepA (RKQGRVEVLYRASWGTVC), PepB (RKQGRVEILYRGSWGTVC), PepC (RKQGRVEVLYRASWGTVS-℗) (℗ = phosphoserine), PepD (RKQGRVEVIYRGSWGTVC), and PepE (RKQGRAEALYRASWGTVC) were supplied by Proteogenix (Schiltigheim, France) with a purity > 85%.

### 2.2. Methods

#### 2.2.1. Synthesis, Functionalization, and Physicochemical Characterization of SPION-APTES

The synthesis and functionalization of SPION-APTES was performed as described recently [18]. Briefly, an aqueous solution of FeCl_2_ and FeCl_3_ (molar ratio 1:2) was stirred and heated to 90 °C under an argon atmosphere. Precipitation was initiated by the addition of 25% ammonia. Afterwards, the obtained particles were stirred at 70 °C for 15 min, followed by the addition of (3-Aminopropyl) triethoxysilane (APTES) to the mixture. The dispersion was stirred for another 3 h before it was cooled to RT. The particles were magnetically separated and washed three times with water. Functionalization was carried out using *N*-Succinimidyl bromoacetate as previously described [18]. A selection of peptides (PepA, PepB and PepE) was then bound to the particles as described. The bound amount of peptide was determined by UV-Vis measurements of the supernatant, after the functionalization step, at a wavelength of 280 nm. The physicochemical properties of the functionalized SPIONs were analyzed regarding their hydrodynamic size, magnetic susceptibility and ζ-potential according to previously published protocols [18,23]. For the determination of the iron content, samples were diluted in deionized H_2_O and liquefied in 65% nitric acid. Analysis was carried out by atomic emission spectroscopy (AES), using the 4200 MP-AES (Agilent Technologies, Santa Clara, CA, USA) with serial dilutions of an iron solution of 1 g/L as an external standard (Bernd Kraft, Duisburg, Germany). Briefly, the functionalized SPIONs were diluted to 50 µg Fe/mL for hydrodynamic size measurements using the Zetasizer Nano ZS (Malvern Panalytical, Almelo, The Netherlands) at 25 °C in water, at a pH of 7.4 (refractive index 1.33, viscosity 0.8872 mPa s, backscattering mode at 173°) and the same device was used to measure the ζ-potential with 78.5 as dielectric constant. The magnetic susceptibility was investigated using a magnetic susceptibility meter (MS2G, Bartington Instruments, Witney, UK) at an iron concentration of 1 mg/mL for all functionalized particles. TEM images were acquired using a Philips CM30 electron microscope (Philips, Netherlands) with an acceleration voltage of 300 kV. SPIONs were diluted to a concentration of 5 µg Fe/mL and 10 µL dropped on carbon-coated copper grids (Plano, Wetzlar, Germany) and dried under air conditions.

#### 2.2.2. Bacteria Culture Conditions and Preparation

Experiments were conducted with *S. aureus* [DSM 346] obtained from the Leibniz Institute DSMZ—German Collection of Microorganisms and Cell Cultures GmbH, Braunschweig, Germany as well as with *S. aureus* (ATCC 29213) and *S. pneumoniae* (ATCC 49619), provided by the strain collection of the Institute of Clinical Microbiology, Immunology and Hygiene, University Hospital Erlangen. For the expansion of *S. aureus* or *S. pneumoniae*, a colony was picked from culture plates and grown overnight in RPMI 1640 medium, supplemented with 0.5% FCS or 10% TSB (Art. 211823), respectively, at 37 °C in 15 mL falcon tubes under aerobic conditions with 5% CO_2_, similar to a protocol reported by Leito et al. [24]. Bacteria were harvested by centrifugation (4500× *g* for 5 min at RT), washed in TBST-Ca (10 mM Tris-HCl pH 7.5, 150 mM sodium chloride, 0.1% Tween 20, 1 mM CaCl_2_) or in Ringer salt solution and resuspended in TBST-Ca or Ringer to an optical density of 1.0 at 600 nm (OD_600_).

#### 2.2.3. Bacterial Agglutination and Turbidometric Assays

The agglutination of Gram-positive bacteria was studied using *S. aureus* (DSM 346 and ATCC 29213) as a model bacterium for evaluating the different peptides, as recently described by Bikker et al. [24] with PepA to PepE. Briefly, 180 µL of the previously adjusted bacterial suspension were added to each well of a 96-well round-bottom microplate (Sarstedt, Nümbrecht, Germany and Greiner Bio-One). An amount of 20 µL of peptide solution was added to reach a final peptide concentration of 0.1 mM. The plates were then incubated at 37 °C. After 30 min, pictures of the bottom of the plates were taken to examine the agglutination. Further, turbidometric analysis of the agglutination process was carried out using a spectrophotometer (OD_600_ DiluPhotometer, Implen GmbH, München, Germany). *S. aureus* and *S. pneumoniae*, adjusted to an OD_600_ of 1.0 as described before [25], were incubated for a period of 2 h at 37 °C after peptides were added under gentle shaking at 250 rpm to prevent sedimentation or auto-agglutination. The optical density of the bacterial suspensions was monitored at 600 nm for 120 min at 37 °C. All experiments were performed in triplicates and PBS was used as a control for both experiments.

#### 2.2.4. Bacteria Separation from Media by Peptide-Functionalized SPION-APTES

Magnetic separation of bacteria was carried out according to a previously published method [18]. The general incubation time was set to 5 min either under flow or shaking conditions, followed by a 3 min magnetic separation at 37 °C. For this study, PepA and PepE/antibPep were used and Ringer solution as a control. After washing, *S. aureus* and *S. pneumoniae* were resuspended in Ringer solution. As medium for the following experiments, Ringer solution and platelet concentrate (only static) were used in bacteria separation experiments under static conditions with a total volume of 1 mL (only shaking at 700 revolutions per minute (rpm)) at a count of approx. 10^3^ CFU/mL and under flow (10 mL/min) in a total volume of 3 mL, using a Mivo^®^ chamber system (React4Life, Genova, Italy) with a bacterial load of 1.0 at OD_600_ (approx. 10^9^ CFU/mL). For separation, the magnet is placed directly beneath the chamber. For experiments with platelet concentrates, the respective samples were obtained from healthy donors from the Department of Transfusion Medicine and Hemostaseology, University Hospital Erlangen. For these tests, bacteria were added as described before. Particles were added to reach a final iron concentration of 300 µg Fe/mL. Magnetic separation was carried out with a neodymium magnet (210 mT, obtained from Webcraft GmbH Gottmadingen, Germany) on the side of the Eppendorf tube (static) or below the separation chamber (flow experiments). Supernatants were either directly analyzed by OD_600_ measurements or, for lower bacterial load (10^3^ CFU/mL), plated out to count the number of bacteria (calculated back to the amount in 1 mL).

#### 2.2.5. Separation Efficiency of LTA by Peptide-Functionalized SPION-APTES

To analyze the separation efficiency of SPION-APTES-PepA, the supernatants were analyzed using a lipoteichoic acid ELISA kit (MyBioSource Inc., San Diego, CA, USA). For separation experiments, we used LTA from *S. aureus* (Sigma-Aldrich Chemie GmbH, Taufkirchen, Germany). The concentrations of the LTA solutions used for separation were between 10 and 1000 ng LTA/mL. Separations were carried out with a particle concentration of 1 µg, 10 µg, or 100 µg Fe/mL of SPION-APTES-PepA. Separations were carried out in Ringer solution as described for separating intact bacteria. Samples were diluted to concentrations that were determined to be appropriate for the respective assays. For the LTA-ELISA, standards were serially diluted from 20 to 0.312 ng/mL. The sample diluent was used as blank. ELISA was carried out according to the manual of the kit. After the addition of the stop solution, the optical density was measured at 450 nm with a SpectraMax iD3 Plate reader (Molecular Devices, San José, CA, USA). Additionally, after separation, particles (100 µg Fe/mL) were analyzed for their ζ-potential as mentioned in Section 2.1.

### 2.3. Statistics

Tests were carried out using one-way or two-way ANOVA, Kruskal–Wallis test and the appropriate post hoc tests. Prior to statistical analysis, the data were tested for normal distribution by Kolmogorov–Smirnov test. Analyses were performed using Prism 9.0.2. The *p* values were two-sided, and asterisks (*) were used to mark statistical significance as relevant for the used test (see figure legend). Any *p* values higher than 0.05 were assumed to be non-significant (ns).

## 3. Results

### 3.1. Agglutination of Gram-Positive Bacteria

To evaluate peptides that can be later used for the binding interface of the iron particles, several pilot experiments were necessary. Turbidometric assays were conducted with the Gram-positive bacteria *S. aureus* and *S. pneumoniae*. An even easier method is the optical evaluation of the agglutination by the addition of peptide to a bacterial suspension. The terminal cysteine in the peptide sequence offers the possibility of ligation via chemoselective linkers to particles. In some experiments, we used a peptide with cysteine exchanged to phosphoserine, which offers the possibility to attach to surfaces such as hydroxylapatite without the necessity of additional linker molecules [26,27].

As shown in Figure 1, the kinetics of the agglutination and the remaining number of different bacteria vary for the different peptides at a concentration of 0.1 µmol peptide/mL. At a lower concentration of 0.01 µmol peptide/mL, no agglutination was observed. In comparison to *S. aureus*, the agglutination of *S. pneumoniae* took more time but resulted in similar OD_600nm_ values for all peptides, except for PepE. Neither control (PBS) nor the PepE (antibPep) showed significant changes in the OD values over time; only a minimal reduction due to the slight bacterial sedimentation was noted. The best agglutination (lowest final OD value after 2 h) for both bacteria tested was observed for PepA and PepD, with a slightly faster agglutination for PepA. PepB and PepC had a minimal lower efficiency, while PepE showed the weakest agglutination for *S. aureus*.

As exemplarily presented in Figure 2 for *S. aureus*, several peptides were capable of binding to bacteria and causing agglutination. The agglutination was visible with the bare eye. In the PBS control and with the peptide PepE, no agglutination occurred.

Previous evaluations of GP-340 peptides have revealed that the amino acid sequence directly affects the binding to pathogens. PepA (RKQGRVEVLYRASWGTVC) and the slightly modified version PepC (RKQGRVEVLYRASWGTVS-℗)) originating from the SRCR2 domain of a sequence tested by Leito et al. that was evaluated to have a higher helical content than other peptides, promoting the binding and agglutination of the peptide [24]. PepB (RKQGRVEILYRGSWGTVC) contains an exchange of valine for isoleucine compared to the binding sequence (xxVEVLxxxxW -> xxVEILxxxxW) originally described by Bikker et al. [28], representing the peptide sequence of the respective part of the human SRCR1 Domain of GP340. PepD (RKQGRVEVIYRGSWGTVC) contains an exchange of leucine for isoleucine (xxVEVLxxxxW -> xxVEVIxxxxW). Both sequence variants did not change the ability to agglutinate the tested bacteria. PepE (RKQGRAEALYRASWGTVC) contains several exchanges of valines for alanines that led to an inability to agglutinate *S. aureus* in the experiments presented (see Figure 1 and Figure 2).

### 3.2. Physicochemical Characterization of Peptide-Functionalized SPIONs

SPION-APTES-Pep have been thoroughly characterized considering SEM, TEM, XRD, and VSM in our previous work [18]. According to this, the particles show superparamagnetic behavior in terms of crystallite size and crystallinity. Further, it was necessary to see if the magnetic driving force determined by the susceptibility did not vary too much to ensure that the separation was still possible at the found timepoints.

For the target-binding experiments, three peptides were selected to be bound to SPION-APTES, as shown in Figure 3. The amount of peptide bound to the SPIONs did not vary for the different peptides.

The physicochemical characteristics of SPION-APTES functionalized with various peptides did not show significant differences, as depicted in Figure 4. The hydrodynamic size (z-average) of the particles was found to be around 1.7 µm with a polydispersity index (PDI) of approx. 0.2–0.3 (Figure 4A,B). It should be considered that the dynamic light scattering DLS measurements show the hydrodynamic size of aggregates formed by small cores of about 12 nm (see also Appendix A Figure A1), as also shown in a previous publication [18]. Further, this study also showed that superparamagnetic behavior was still visible after the functionalization with peptide, as seen in the VSM measurements.

The ζ-potential was found to be 29 mV for all functionalized particles (Figure 4C). All particles exhibited a good visible magnetic removability from media, which is a prerequisite for the capture of bacteria. As reported before in previous studies, the particles proved to have good bio- and hemocompatibility, making them usable for separation under physiological conditions [18].

### 3.3. Removal of Gram-Positive Bacteria from Media under Static and Flow Conditions by Peptide-Functionalized SPION-APTES

Next, the SPION-APTES particles functionalized with different peptides were tested for the magnetic removal of intact bacteria from the suspensions. Ringer solution was chosen as the aqueous medium because it contains Ca^2+^, which is necessary for the binding of the peptide to the bacteria [25,29]. Under static conditions, SPION-APTES PepE/antibPep failed to bind and remove the tested bacteria from the medium (Figure 5), confirming the results from the agglutination experiments (see Figure 1). A slight difference between the three peptides (A, B, and E) and the two bacterial species tested was found, as before. Higher amounts of *S. pneumoniae* remained in the supernatant compared to *S. aureus*. SPION-APTES-PepA could almost completely remove *S. aureus* from the medium, while a minor amount of *S. pneumoniae* remained in the supernatant after separation. Efficiency of SPION- APTES-PepA was over 99% and about 96% for removal of *S. aureus* and *S. pneumoniae*, respectively. For SPION-APTES-PepB, these values were about 98% and still about 91%, respectively.

Separations that were further performed under a constant flow of 10 mL/min after 5 min of incubation (during constant flow) with SPIONs with PepA and PepE (see the device as presented in Figure 6) revealed that about 70% of *S. aureus* (high load 10^9^ CFU/mL OD_600_ 1.0) were removed from the medium if the particles were functionalized with PepA, while particles functionalized with PepE failed to do so (Figure 6).

### 3.4. Removal of LTA by Peptide-Functionalized SPION-APTES

For these experiments, PepA and, as a control, PepE (anti-binding peptide) were selected, as presented in Section 3.1 and Section 3.3, and bound to SPION-APTES. Different concentrations of SPION-APTES-PepA were used for the separation of 10 ng LTA/mL (Figure 7A) and 1000 ng LTA/mL (Figure 7B). It was found that 100 µg Fe/mL of SPION-APTES-PepA was able to completely remove LTA at both concentrations from the used medium after 5 min of incubation followed by 3 min of magnetic separation. In contrast, SPION-APTES-PepA at 1 or 10 µg Fe/mL were only partially able to remove LTA from the medium or showed no significant decrease.

To prove further the binding of LTA onto the particles (at a concentration of 100 µg Fe/mL), they were washed after separation of 10, 1000, and 10,000 ng/mL LTA and the ζ-potential was analyzed afterwards. The results presented in Figure 8 show a drop in the ζ-potential depending on the amount of LTA, confirming the successful binding of LTA onto the particles. It was further confirmed that no binding to particles with PepE (anti-binding peptide) occurred, as no significant drop in ζ-potential was found.

### 3.5. Removal and Enrichment of Separated Bacteria from Platelet Concentrate

A potential application of the peptide-functionalized SPIONs is the separation of *S. aureus* from platelet concentrates. As mentioned before, bacterial contamination of these concentrates can cause serious infections with fatal outcomes [30]. As shown in Figure 9, SPION-APTES-PepA were able to remove 63% of the bacteria after one separation cycle. The amount of concentrated, particle-bound bacteria after the separation was increased by a factor of four compared with the untreated sample.

## 4. Discussion

Several peptide sequences derived from GP-340 were evaluated for their ability to agglutinate the Gram-positive bacteria *S. aureus* and *S. pneumoniae*. We found that those peptides were able to agglutinate the investigated bacteria to different extents (macroscopically visible after 30 min). We confirmed that an exchange of valines to alanines in the peptide sequence comprising the proposed minimal binding motive (xxVEVLxxxxW -> xxAEALxxxxW) led to the loss of the ability to bind to the tested bacteria. Quantitative differences in the agglutination behavior for the different peptides are possibly due to conformational changes or refolding processes, which will affect the affinity properties of the peptide.

The structure and composition of the cell wall of the targeted bacteria are important for their binding to the peptide-functionalized bacteria. However, the variances in the cell walls of the two tested bacteria minimally influenced the binding efficiency, as seen in the bacterial separation experiments with the peptide-functionalized particles. The cell wall of *S. pneumoniae* has complex repeating units in the lipoteichoic acids that can lead to more or different peptide attachments to the walls of these bacteria and make longer incubation times necessary [11]. In this respect, it is likely that the number of bacteria that can be crosslinked is reduced and thereby the agglutination is less strong. Recent studies have shown that the amounts of the added peptide can influence agglutination [25,29]. Therefore, higher amounts of peptide need to be considered in the future as well. Furthermore, LTA varies between different Gram-positive bacteria or with the serotype of the bacteria and might possess different affinities to the GP-340-derived peptides, resulting in a higher or weaker agglutination [31,32,33].

The results from the agglutination were successfully translated to the separation experiments with particle-bound peptides, whereas a direct screening of particle-bound peptides would be much more time- and material-consuming. The different peptides were bound to SPION-APTES using a well-established procedure. The peptide-functionalized particles were capable of removing high and clinically relevant quantities of bacteria [34] from an aqueous medium. Separation occurred within 5 min of co-incubation and was therefore faster and more efficient than agglutination with the bare peptide, probably because of the larger surface area and interaction of the particles with the bacteria [35,36]. Another reason could be that, by using the peptides alone, one peptide has to interconnect at least two or more bacteria to start agglutination, but by using particle-bound peptides the agglutination could be based on multiple peptides with the SPION acting as an interconnection point. Besides the direct removal of the bacteria, free LTA, the target structure of the peptides, was also efficiently removed using the peptide-functionalized SPIONs. A dose-dependent removal was observed, showing that it was possible to remove 1 µg LTA/mL using SPIONs at a concentration of 100 µg Fe/mL. Interestingly, at higher concentrations of LTA (from *S. aureus*) a particle amount of 100 µg Fe/mL could remove a higher proportion of LTA than at 10 ng LTA/mL. This is likely due to the fact that the LTA and the particles will have more interactions if the concentration of the interaction partner is higher. Furthermore, it is possible that LTA at high concentrations can form micelles [37,38,39] that might be removed more efficiently than monomeric LTA. As expected from the agglutination experiments, the PepE (anti-binding peptide) failed to remove LTA even at a concentration of 100 µg Fe/mL.

The successful binding of the particles to LTA was further proven by the analysis of the ζ-potential after the separation procedure. The potential went from +29 mV to around −30 mV (for 10 µg LTA/mL), proving the successful binding to the particles as LTA exhibits negative charges [40]. Experiments using a flow chamber demonstrated that the removal of LTA is possible even under conditions of a permanent flow of 10 mL/min without significant reduction in efficacy. Such a system can be used to evaluate whether LTA can be removed from larger volumes, as it would be required for therapeutic applications.

In a final approach, *S. aureus* was removed from platelet concentrates to a degree of more than 63% after one single cycle of separation. The captured bacteria could be re-cultured from and due to the reduction of dispersion volume of the particle-bacteria-complex. A four times higher concentration of CFU was found than from the unseparated bacterial suspension (the volume used for plating out the samples was 50 µL). A lower total separation rate compared with the water-based system was previously observed in separation experiments with total blood [18]. The presence of blood cells, but also anticoagulants, might account for the reduced efficiency. However, this technique nevertheless offers a relevant improvement that could be used for, e.g., a faster diagnosis of infections of the bloodstream or blood products. Furthermore, compared to antibodies against bacteria that possess a narrow target spectrum [41], the used peptides exhibit a far broader binding spectrum.

Further studies and simulations are needed to elucidate structure–activity relationships between peptide and target at the molecular mechanistic level and possibly predict affinities for specific bacterial species. Moreover, it has to be determined whether, under certain conditions, the peptides form larger aggregates in media that are responsible for the strong agglutination behavior.

## 5. Conclusions

In summary, the presented results of this study demonstrate that certain GP-340-derived peptides are able to agglutinate the Gram-positive bacteria *S. aureus* and *S. pneumoniae*. Furthermore, it was shown that distinct amino acid exchanges in the peptide sequence can lead to a loss of the ability to agglutinate bacteria. The combination of SPION-APTES and peptide (SPION-APTES-Pep) can be synthesized reliably with different peptides as long as they possess a C-terminal cysteine. Particle-bound peptides were able to remove different Gram-positive bacteria and/or LTA from aqueous medium with high efficiency, even under flow conditions. In first experiments, we showed that peptide-functionalized particles successfully enriched *S. aureus* from platelet concentrates, which opens up a low-threshold field of application. SPION-APTES-Pep offer an efficient procedure for diagnosing bloodstream infections and blood product contamination with Gram-positive bacteria and their LTA toxins.

## 6. Patents

The concept of using salivary peptides bound to SPIONs is currently being patented.

## Figures and Tables

**Figure 1 pharmaceutics-14-02356-f001:**
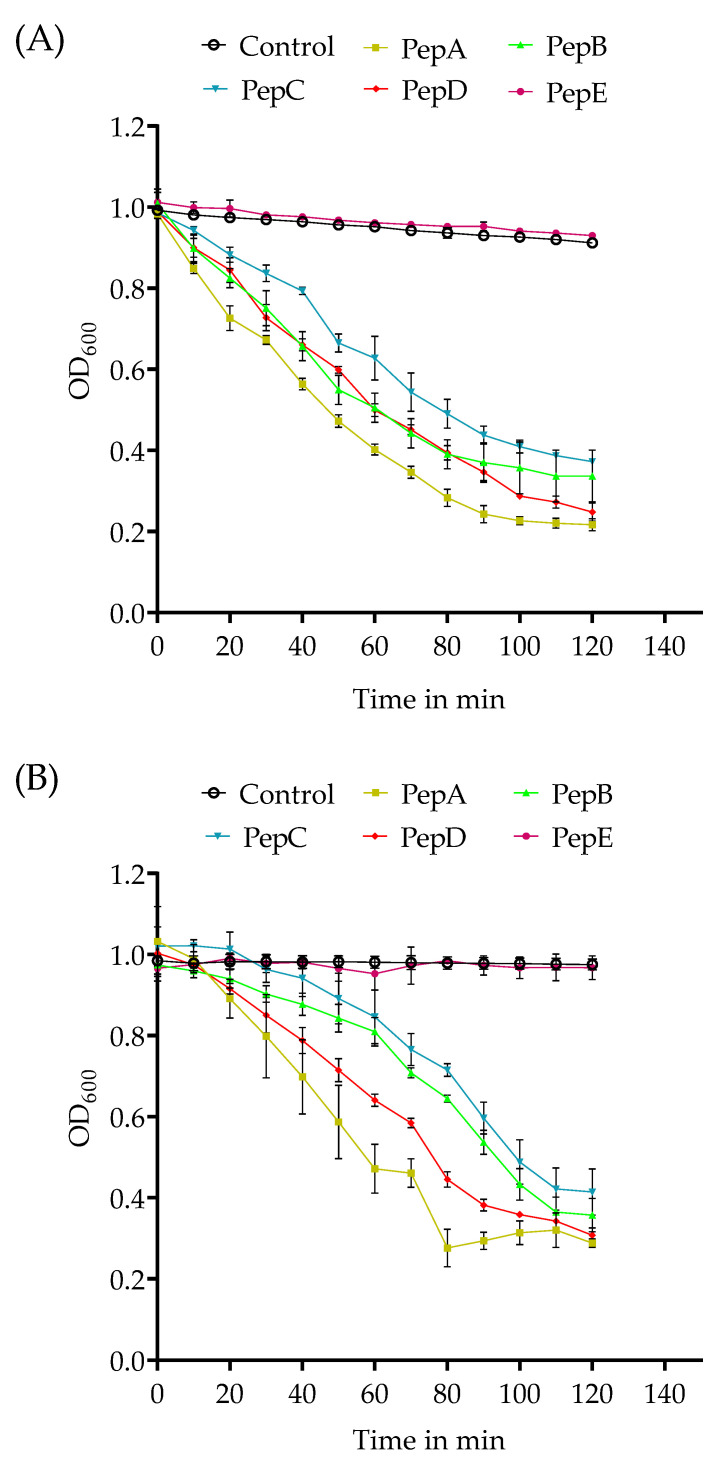
Turbidometric agglutination of *S. aureus* (panel (**A**)) and *S. pneumoniae* (panel (**B**)) after incubation for 2 h at 37 °C with different peptides (0.1 µmol peptide/mL). PBS served as the control. The results shown represent the mean values with standard deviations of three independent experiments.

**Figure 2 pharmaceutics-14-02356-f002:**
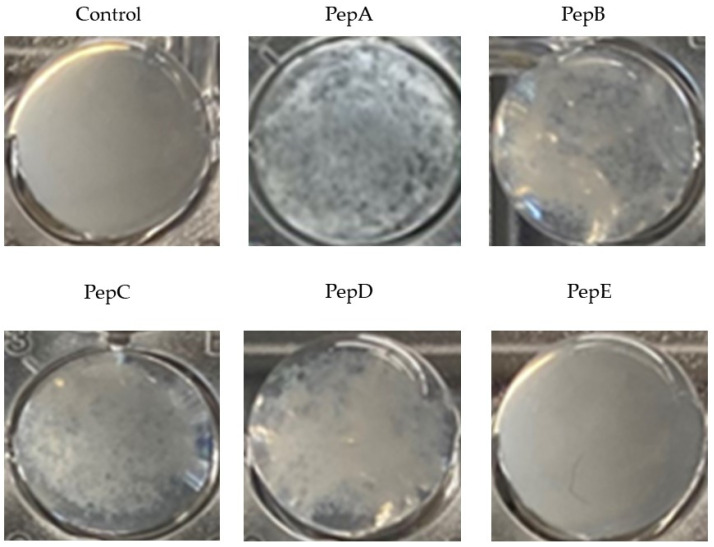
Macroscopic appearance of the GP340-peptide-mediated agglutination of *S. aureus* shown for different peptides at a concentration of 0.1 µmol/mL and PBS as control after incubations for 30 min at 37 °C in 96-well plates.

**Figure 3 pharmaceutics-14-02356-f003:**
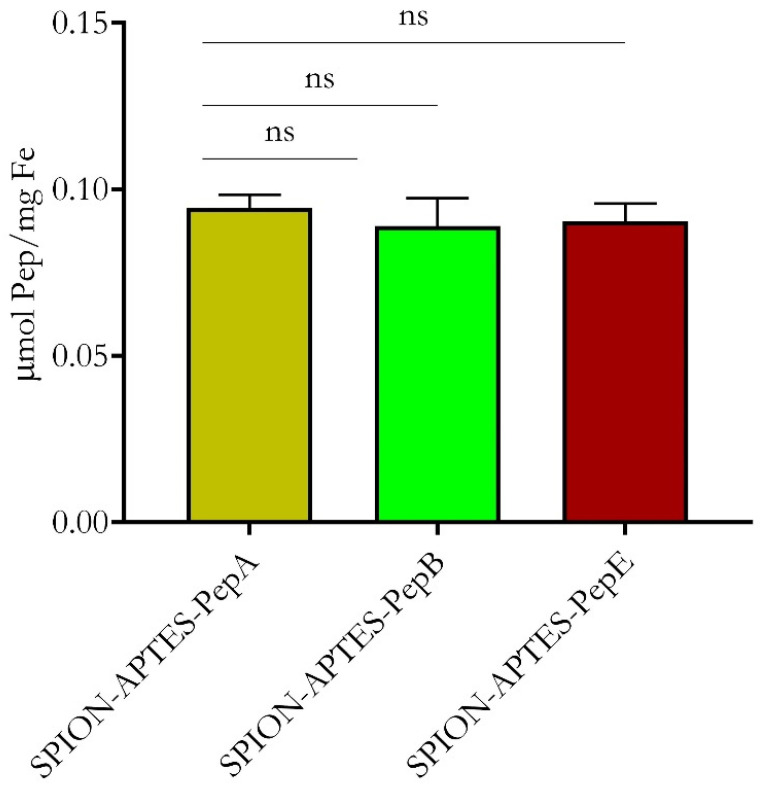
The amount of different peptides bound to SPION-APTES as determined by measurement of supernatants at 280 nm after the functionalization procedure. The results shown represent the mean values with standard deviations of three independent experiments. Statistical testing was performed using a Kruskal–Wallis test. *p* values above 0.05 were considered non-significant (ns).

**Figure 4 pharmaceutics-14-02356-f004:**
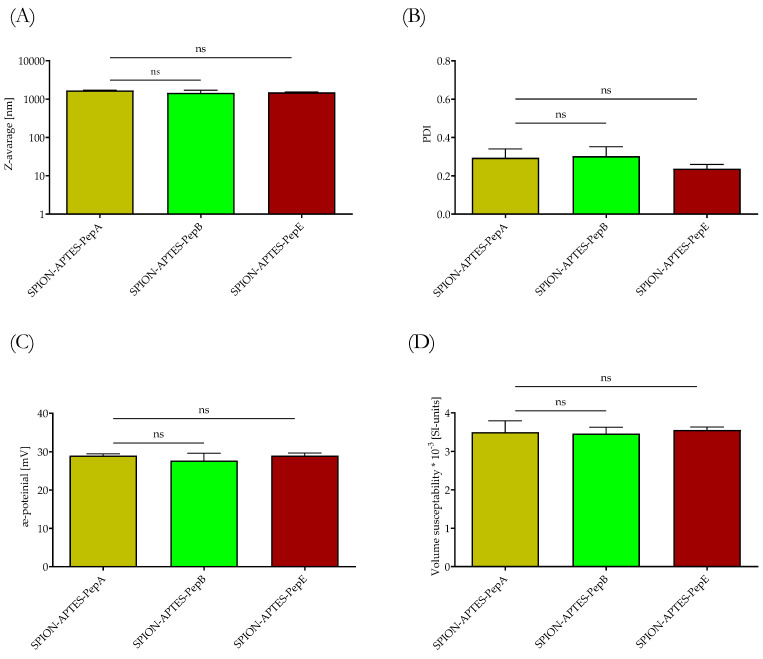
(**A**) Hydrodynamic size, (**B**) PDI, (**C**) ζ-potential, and (**D**) volume susceptibility of the different peptides bound to SPION-APTES. Shown are the mean values of standard deviations. Statistical testing was performed using a one-way ANOVA test. *p* values above 0.05 were considered non-significant (ns).

**Figure 5 pharmaceutics-14-02356-f005:**
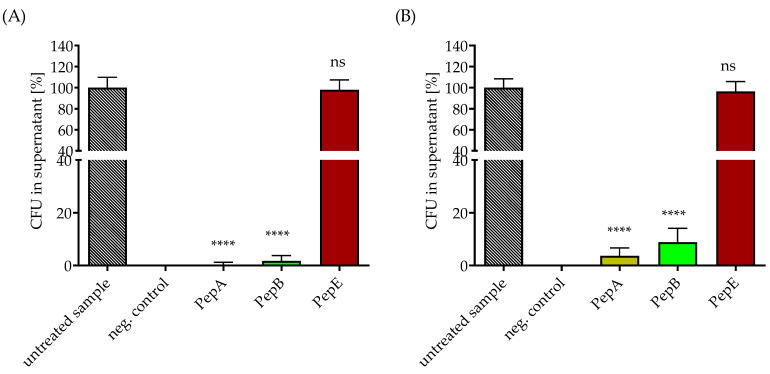
Removal of *S. aureus* (panel (**A**)) or *S. pneumoniae* (panel (**B**)) under static conditions from Ringer solution. The concentration of bacteria was set to 10^3^ CFU/mL. Separations were carried out with 300 µg Fe/mL peptide-functionalized SPION-APTES. Unspiked samples served as negative control. The results shown represent the mean values with standard deviations of three independent experiments. Statistical testing was performed between untreated and treated samples using a two-way ANOVA test with a post hoc Dunn’s test; **** *p* < 0.0001. *p* values above 0.05 were considered non-significant (ns).

**Figure 6 pharmaceutics-14-02356-f006:**
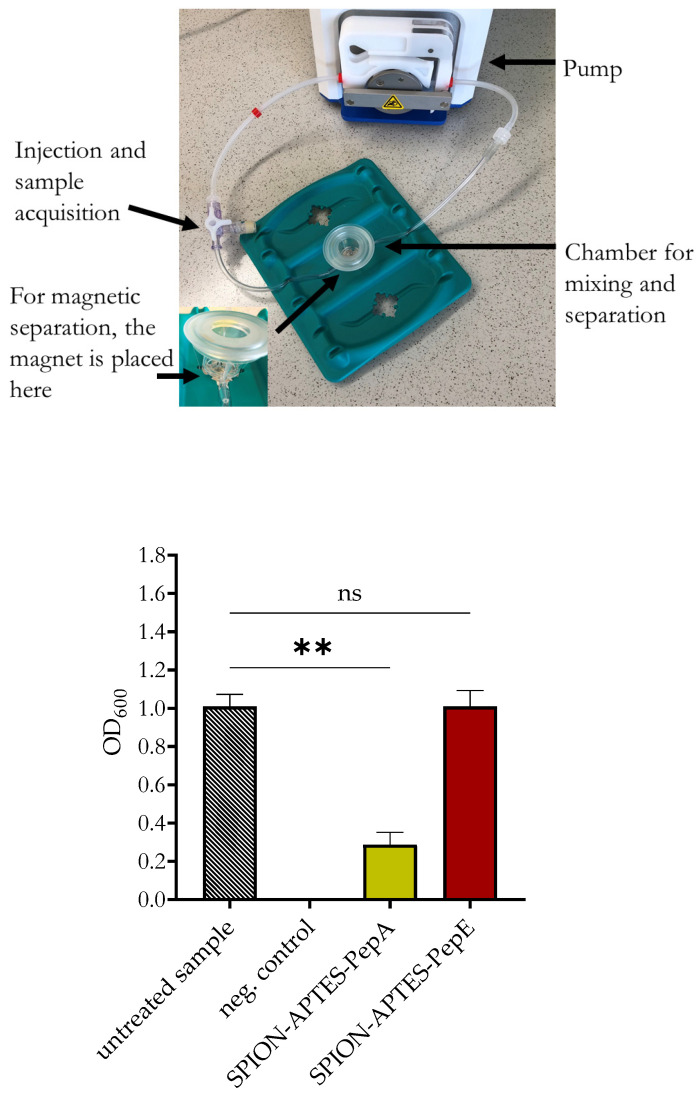
Above: Separation device for flow experiments. Below: Removal of *S. aureus* under flow conditions (10 mL/min, see also Figure A1) in Ringer solution using SPION-APTES-PepA and SPION-APTES-PepE at a concentration of 300 µg Fe/mL. The results shown represent the mean values with standard deviations of three independent experiments. Statistical testing was performed between untreated and treated samples using a Kruskal–Wallis test with a post hoc Dunn’s test; ** *p* ≤ 0.05. *p* values above 0.05 were considered non-significant (ns).

**Figure 7 pharmaceutics-14-02356-f007:**
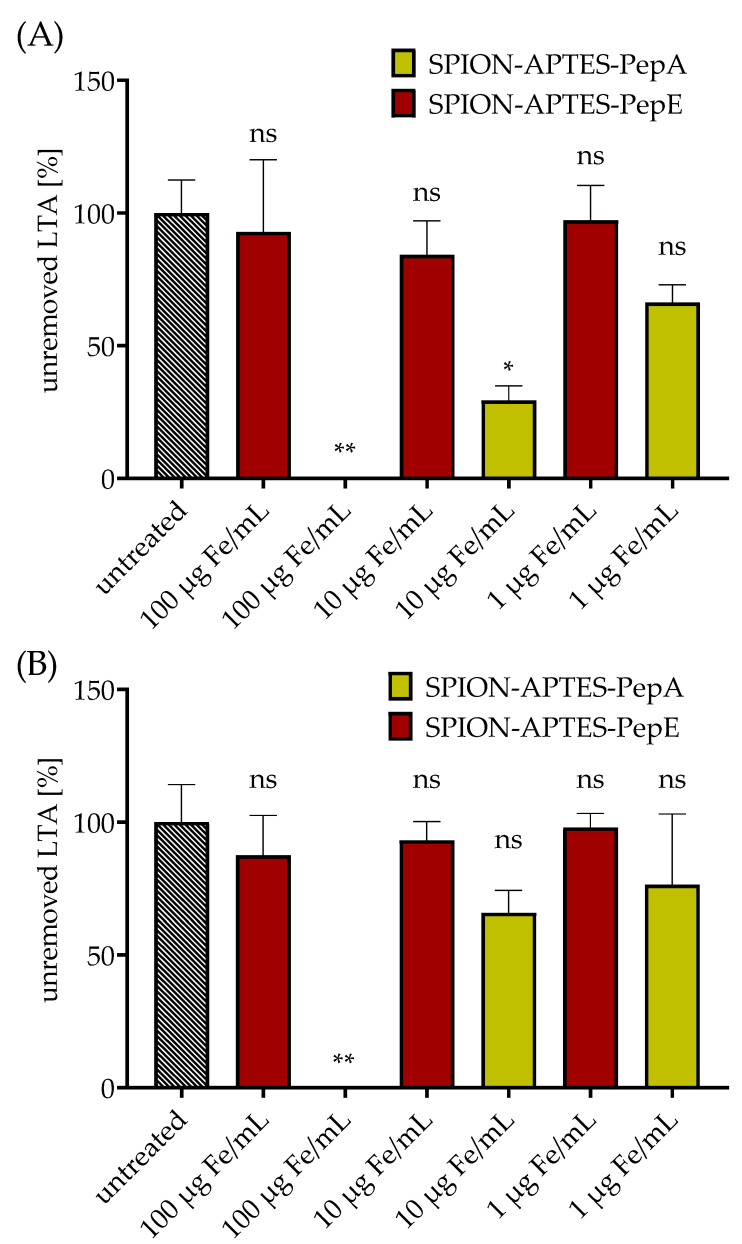
Removal of S. aureus LTA from water-based media. Separations were carried out with LTA concentrations of 10 ng LTA/mL (panel (**A**)) and 1000 ng LTA/mL (panel (**B**)). SPION-APTES-PepA and SPION-APTES-PepE were used at concentrations of 1, 10, or 100 µg Fe/mL. The results shown represent the mean values with standard deviations of three independent experiments. Statistical testing was performed between untreated and treated samples using a Kruskal–Wallis test with a post hoc Dunn’s test; * *p* ≤ 0.05; ** *p* ≤ 0.01. *p* values above 0.05 were considered non-significant (ns).

**Figure 8 pharmaceutics-14-02356-f008:**
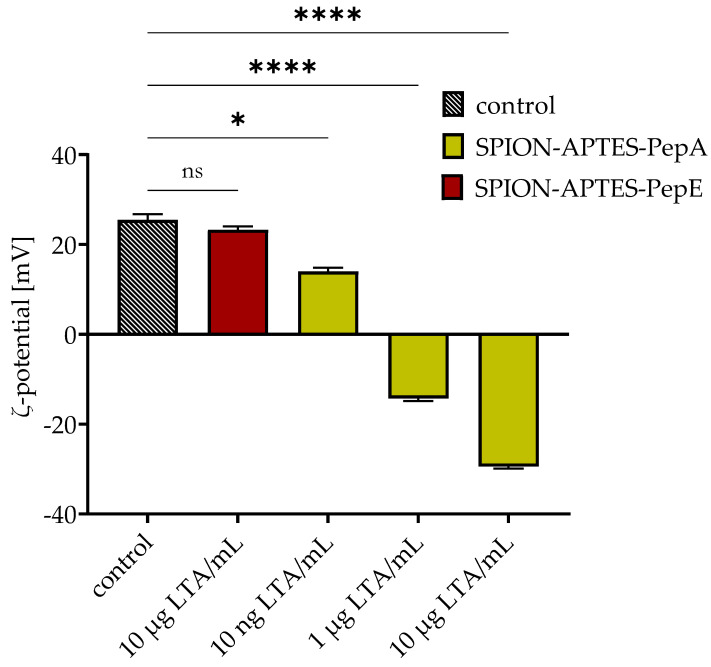
Changes in ζ-potential after separation of different amounts of LTA (10, 1000, and 10,000 ng LTA/mL) using 300 µg Fe/mL of SPION-APTES-PepA and SPION-APTES-PepE. SPION-APTES-PepA without added LTA during separation served as control. The results shown represent the mean values with standard deviations of three independent experiments. Statistical testing was performed using a Kruskal–Wallis test with a post hoc Dunn’s test; * *p* ≤ 0.05; **** *p* < 0.0001. *p* values above 0.05 were considered non-significant (ns).

**Figure 9 pharmaceutics-14-02356-f009:**
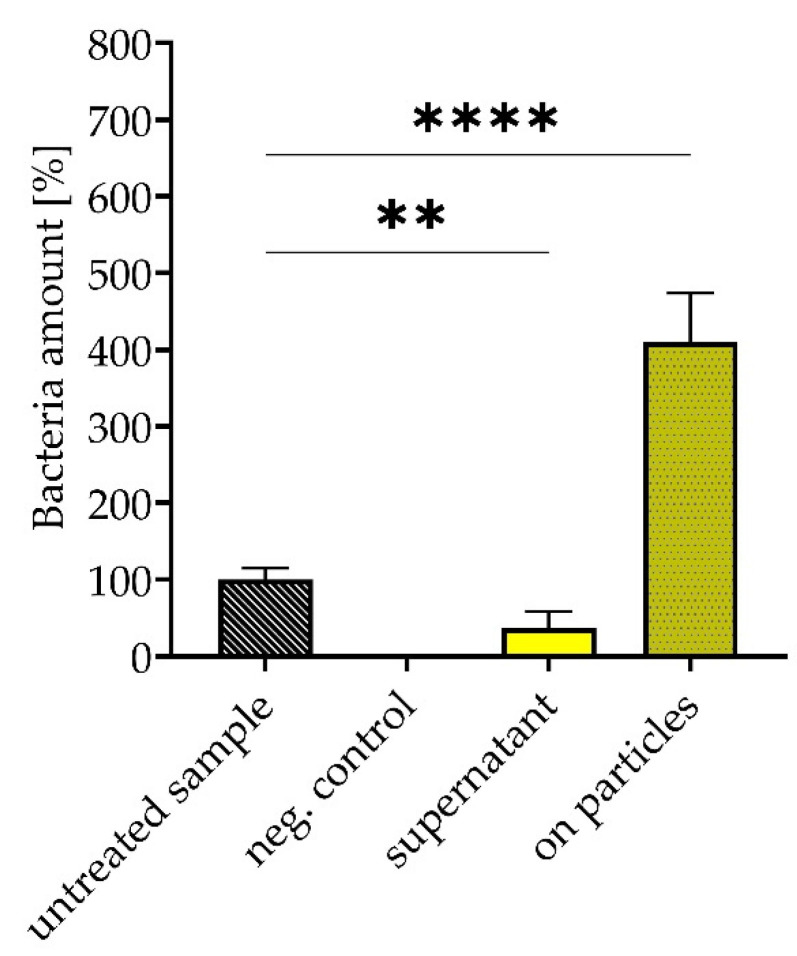
Removal and enrichment of *S. aureus* from platelet concentrate using 300 µg Fe/mL SPION-APTES-PepA. Samples were spiked with bacteria at a concentration of 10^3^ CFU/mL. The unspiked platelet concentrate served as negative control. Experiments were performed in triplicates. The results shown represent the mean values with standard deviations of three independent donors. Statistical testing was performed using a two-way ANOVA test with post Dunn’s test; ** *p* ≤ 0.01; **** *p* < 0.0001. *p* values above 0.05 were considered non-significant.

## Data Availability

Data is available from the authors by reasonable request.

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
