# Peer review of "Biomimetic Magnetic Particles for the Removal of Gram-Positive Bacteria and Lipoteichoic Acid"

_pharmaceutics, 2022, doi:10.3390/pharmaceutics14112356_

Round 1

Reviewer 1 Report

In this work, Friedrich and co-workers described an interesting application of magnetic particles decorated with different peptides to bind specific types of bacteria, Gram-positive ones, and lipoteichoic acid. The experiments were well designed and reasonably presented. Statistical analysis was fine. Control experiments were sufficient. The conclusions were strongly based on experimental data. The research topic fits the scope of this Journal well. Therefore, I would think this work is in very good shape for publication.

I would recommend the authors to perform a minor revision on the manuscript. The following are just a few examples:

            Line 23: please spell out as “For example,”;

            Line 24: extra comma;

Line 34: missing the first half of the parenthesis;

            Line 35: the abbreviation should appear the first time, on Line 30;

            Line 80: should be “… consists of only…”;

            Line 193: “rpm” should be spelled out as “revolution per minute” for the first time;

            Line 554: “es” should be “as”.

One more comment is that Figure A1 might be better integrated with the main figure, as Figure 6A. That way the readers will follow a bit more easily.

Author Response

In this work, Friedrich and co-workers described an interesting application of magnetic particles decorated with different peptides to bind specific types of bacteria, Gram-positive ones, and lipoteichoic acid. The experiments were well designed and reasonably presented. Statistical analysis was fine. Control experiments were sufficient. The conclusions were strongly based on experimental data. The research topic fits the scope of this Journal well. Therefore, I would think this work is in very good shape for publication.

We thank the reviewer for his evaluation and of course we address in the newer version his comments. We rectified spelling and grammar mistakes and checked the manuscript again to improve the language. All those changes were highlighted in green.

I would recommend the authors to perform a minor revision on the manuscript. The following are just a few examples:

            Line 23: please spell out as “For example,”;

            Line 24: extra comma;

Line 34: missing the first half of the parenthesis;

            Line 35: the abbreviation should appear the first time, on Line 30;

            Line 80: should be “… consists of only…”;

            Line 193: “rpm” should be spelled out as “revolution per minute” for the first time;

            …

            Line 554: “es” should be “as”.

One more comment is that Figure A1 might be better integrated with the main figure, as Figure 6A. That way the readers will follow a bit more easily.

The reviewer is right. We shifted this figure accordingly to Figure 6

Reviewer 2 Report

What was the size of the magnetic nanoparticles used in the research? What technique was used to measure their size ?

How was the value of their magnetism determined? Could you also short present t how gram-negative bacteria behave under the presented measurement conditions?

Please, clearly indicate the novelty of your study in the introduction.

Author Response

We thank the reviewer for the specific comments, which we would like to address below:

What was the size of the magnetic nanoparticles used in the research? What technique was used to measure their size?

Hydrodynamic size was measured using the Zetasizer Nano ZS (Malvern Panalytical, Almelo, Netherlands). A respective paragraph has been added to the methods part (chapter 2.2.1):

Briefly the functionalized SPIONs were diluted to 50 µg Fe/mL for hydrodynamic size measurements using the Zetasizer Nano ZS (Malvern Panalytical, Almelo, Netherlands) at 25 °C in water at a pH of 7.4 (refractive index 1.33, viscosity 0.8872 mPa s, backscattering mode at 173°) and ζ-potential the same device was used to measure the zeta potential with 78.5 as dielectric constant.

Further determination of the size of the cores was previously carried out in our recent publication  (https://doi.org/10.1016/j.actbio.2022.01.001) in which material properties have been characterized thoroughly. We refer to this elsewhere in the manuscript in chapter 3.2:

SPION-APTES-Pep have been thoroughly characterized considering SEM, TEM, XRD and VSM in our previous work [29]. According to this, the particles show super paramagnetic behavior in terms of crystallite size and crystallinity.

How was the value of their magnetism determined? Could you also short present how gram-negative bacteria behave under the presented measurement conditions?

A paragraph for the magnetic measurements has been added to the Methods part (Chapter 2.2.1)

The magnetic susceptibility was investigated using a magnetic susceptibility meter (MS2G, Bartington Instruments, Witney, UK) at an iron concentration of 1 mg/mL for all functionalized particles.

VSM analysis which is a very suitably method for determining magnetic properties was provided in the recent publication (https://doi.org/10.1016/j.actbio.2022.01.001). In that study we focused on gram negative bacteria and LPS which is the bacterial product of those species. Briefly, they can also be removed very efficiently, since the recognition pattern for the peptide is phosphate saccharides, which is inherent to both LPS and LTA. Magnetic measurements have not been performed with bacteria in addition as the aim of this study was to see if the bacteria can be removed from media not the changes to the magnetic properties of bacteria-SPION dispersions.

Please, clearly indicate the novelty of your study in the introduction.

We have rearranged the last paragraph of the introduction at several places to make the novelty clearer (highlighted in green). We wanted to describe that for the first time we have carried out particle-bound binding experiments with Gram-positive bacteria and LTA, especially at relevant concentrations. We were also able to make the findings directly applicable to a specific medical problem, contamination of blood products.

Reviewer 3 Report

The aim of this paper is clearly stated and it’s a topic of great interest for applications. This paper focus on the synthesis, characterization and applications of functionalized superparamagnetic iron oxide particles with attached peptides derived from the innate immune receptor GP-340 to bind and remove Gram-positive bacteria and lipoteichoic acid (LTA) from different media.  The authors have done a careful analysis of the ability to agglutinate the Gram-positive bacteria S. aureus and S. pneumoniae for nanoparticles functionalized with several peptide sequences derived from GP-340.

The paper contains interesting results and could have impact in the field of biomedical applications of functionalized iron oxide nanoparticles.  

I recommend the publication of this paper.

Author Response

We thank the reviewer for his very encouraging evaluation.

Reviewer 4 Report

The manuscript is not written well. However, there are many major issues that should be clarified or corrected by the authors:

1.      Authors must describe a clear-cut novelty, objective and major findings of the present work in abstract.

2.      In introduction section authors should described about antimicrobial and antibiofilm potential of magnetic nanoparticles.

3.      What are different characterization techniques used in this study to characterized the synthesized SPION, described in detailed I methodology section.

4.      SEM, TEM, XRD, FTIR, VSM are completely missing.

5.      What was the size and crystalline nature of SPION.

6.      How the size of the nanoparticles has been calculated.

7.      How authors described that the synthesized magnetic nanoparticles were super para magnetic in nature, without conducting in experiment related to magnetic properties characterization techniques.

8.      In this current form it is not suitable for publication.

Author Response

The manuscript is not written well. However, there are many major issues that should be clarified or corrected by the authors:

We thank the reviewer for his critical evaluation of our work and address his comments accordingly which would further improve our manuscript.

  1. Authors must describe a clear-cut novelty, objective and major findings of the present work in abstract.

In the abstract our key fact was already mentioned that, we evaluated superparamagnetic iron oxide particles (SPION) carrying peptides derived from the innate immune receptor (GP-340). “For this purpose, we took advantage on a very efficient biomimetic pathogen immobilization strategy using peptide fragments derived from the agglutinating salivary protein (GP-340).” Maybe this was not sound enough so we changed the formulation at several points (highlighted in green):

For this purpose, we took advantage on a very efficient biomimetic pathogen immobilization strategy using peptide fragments derived from the agglutinating salivary protein (GP-340).

Other convincing arguments, such as separation under flow conditions and testing with a direct application reference (detection of microbial contaminants in blood products) are clearly presented now.

  1. In introduction section authors should described about antimicrobial and antibiofilm potential of magnetic nanoparticles.

The aim of our work was not to disintegrate bacteria or biofilms but immobilizing pathogens for diagnostic purposes which is a completely different approach. Therefore, antimicrobial and antibiofilm potential was not mentioned in the introduction. An investigation of the interaction of our particles with biofilms could be an interesting study in the future but was not the goal of this manuscript.

  1. What are different characterization techniques used in this study to characterized the synthesized SPION, described in detailed I methodology section.

A paragraph (Green) has been added to the methods part in Chapter 2.2.1.

Briefly the functionalized SPIONs were diluted to 50 µg Fe/mL for hydrodynamic size measurements using the Zetasizer Nano ZS (Malvern Panalytical, Almelo, Netherlands) at 25 °C in water at a pH of 7.4 (refrac-tive index 1.33, viscosity 0.8872 mPa s, backscattering mode at 173°) and ζ-potential the same device was used to measure the ζ-potential with 78.5 as dielectric constant. The magnetic susceptibility was investigated using a magnetic susceptibility meter (MS2G, Bartington Instruments, Witney, UK) at an iron concentration of 1 mg/mL for all functionalized particles. TEM images were acquired using a Philips CM30 electron microscope (Philips, Netherlands) with an acceleration voltage of 300 kV, providing a point resolution of 0.23 nm. SPIONs were diluted to a concentration of 5 µg Fe/mL and 10 µL dropped on carbon-coated copper grids (Plano, Wetzlar, Germany) and dried under air conditions.

  1. SEM, TEM, XRD, FTIR, VSM are completely missing.

The reviewer is correct, readers can expect appropriate characterizations of such a powerful particle system. In this work, we did not aim to develop a new particle system, but to test SPIONS from our previous work in a new very relevant setting. So not to what extent Gram negative bacteria and LPS can be separated as a bacterial product, but how it behaves with different Gram positive bacteria and LTA as an analogous bacterial toxin. Moreover, we wanted to demonstrate the feasibility of the procedure in an application-oriented setting for detecting microbial contaminants in blood products. In this respect, all physical investigations were already carried out in the previous work (https://doi.org/10.1016/j.actbio.2022.01.001). There, the corresponding investigations like SEM, TEM, XRD, FTIR, VSM were conducted. We will state this in chapter 3.3 in our manuscript as follows:

SPION-APTES-Pep have been thoroughly characterized considering SEM, TEM, XRD and VSM in our previous work [29]. According to this, the particles show super paramagnetic behavior in terms of crystallite size and crystallinity. Further it was necessary to see if the magnetic driving force determined by the susceptibility did not vary too much to ensure that the separation was still possible at the found timepoints.

  1. What was the size and crystalline nature of SPION.

Please take a look at response to your comment no 4.

  1. How the size of the nanoparticles has been calculated.

Hydrodynamic size was measured using the Zetasizer Nano ZS (Malvern Panalytical, Almelo, Netherlands). A paragraph (Green) has been added to the Methods part. Further determination of the size of the cores was previously carried out in our recent publication (https://doi.org/10.1016/j.actbio.2022.01.001).

  1. How authors described that the synthesized magnetic nanoparticles were super para magnetic in nature, without conducting in experiment related to magnetic properties characterization techniques.

VSM measurements were conducted in the mentioned previous publication (https://doi.org/10.1016/j.actbio.2022.01.001)  showing the superparamagnetic nature of the unfunctionalized and the functionalized SPIONs. As the functionalization of was the same for all peptides it can be assumed that the exchange of the peptide is not influencing this behavior.

  1. In this current form it is not suitable for publication.

We hope that our careful revision of the manuscript was able to eliminate the concerns.

Round 2

Reviewer 2 Report

I accept this paper in present form.

Reviewer 4 Report

accept in current form as authors has added all the requested information.